# Factors Associated with Combination Therapy Involving Traditional Korean Medicine in Pediatric Allergic Rhinitis Patients: A Retrospective Cohort Study Using National Health Insurance Data

**DOI:** 10.3390/healthcare13080875

**Published:** 2025-04-11

**Authors:** Mi Ju Son, Kyung-Min Shin, Changsop Yang, Eunkyoung Ahn

**Affiliations:** 1KM Science Research Division, Korea Institute of Oriental Medicine, Daejeon 34054, Republic of Korea; mj714@kiom.re.kr (M.J.S.); yangunja@kiom.re.kr (C.Y.); 2KM Data Division, Korea Institute of Oriental Medicine, Daejeon 34054, Republic of Korea

**Keywords:** pediatrics, allergic rhinitis, traditional Korean medicine, National Health Insurance data

## Abstract

**Background/Objectives**: Allergic rhinitis (AR) significantly affects patients’ quality of life and poses a substantial burden on healthcare systems worldwide, with its prevalence rising among pediatric populations. This study aimed to investigate the factors associated with the use of combination therapy involving traditional Korean medicine (KM) and conventional medicine in pediatric AR patients using large-scale health insurance data from South Korea. **Methods**: Data from 696,182 pediatric patients diagnosed with AR between 2018 and 2021 were analyzed. Multivariate logistic regression was employed to identify factors influencing KM therapy utilization. **Results**: Among these, 61,745 patients received KM combination therapy. The key predictors of KM combination therapy utilization included school-age children, the winter season, and comorbidities such as rhinitis and atopic dermatitis. Patients receiving KM combination therapy had longer treatment durations and incurred higher healthcare costs compared to those on conventional therapy alone. Additionally, seasonal variations and demographic factors, including a decline in KM use during the Coronavirus Disease 2019 (COVID-19) pandemic, were observed. Conclusions: The findings suggest that integrating KM into pediatric AR treatment may offer potential benefits, especially for children with comorbidities or during the winter months. These insights could guide the development of targeted healthcare policies and strategies to optimize treatment outcomes for pediatric AR patients.

## 1. Introduction

Allergic rhinitis (AR) is a common immunological disorder characterized by nasal congestion, rhinorrhea, nasal itching, and sneezing, often accompanied by ocular and pharyngeal symptoms [1,2]. These symptoms can significantly disrupt sleep, impair daytime concentration, and reduce overall quality of life and productivity [3]. Globally, AR affects 10–30% of the population, with approximately 500 million individuals affected as of 2021 [2,4]. In South Korea, the prevalence of AR has steadily increased, with data from the National Health Insurance Service indicating that around 7.03 million individuals sought treatment for AR in 2018, reflecting an average annual growth of 2.6% since 2014 [5]. During the same period, medical expenses for AR treatment reached USD 345 million (KRW 512.7 billion), with an average annual increase of 6.6% [5].

Traditional Korean medicine (KM) plays an important role in managing AR, with its use steadily increasing in recent years. Between 2010 and 2016, the number of AR patients treated in KM settings, along with associated medical claims, rose by approximately 1.4 times, while annual medical expenditures increased by 1.7 times [6]. KM approaches to AR treatment often include herbal medicines, acupuncture, and herb-based nasal therapies [7]. Numerous studies have demonstrated the potential of KM interventions for AR management. Frequently used herbal medicines, such as Socheongryongtang, Okbyeongpoongsan, and Hyeonggaeyeongyotang, have been shown to alleviate symptoms, improve quality of life, and enhance the body’s overall resilience [8,9,10,11]. Certain herbal formulas have been shown to modulate immune responses by reducing serum immunoglobulin E (IgE) levels, suppressing inflammatory mediators, and restoring the balance between Th1 and Th2 cell activities, ultimately alleviating allergic inflammation [12]. Acupuncture, another cornerstone of KM, has demonstrated effectiveness in improving AR symptoms and quality of life, with clinical studies highlighting its benefits compared to placebo treatments [13,14]. Acupuncture exerts therapeutic effects by modulating neuroimmune responses, balancing pro-inflammatory and anti-inflammatory cytokines, and reducing allergen-specific IgE levels and inflammatory neuropeptides, thus effectively improving clinical symptoms and quality of life in patients with AR [15]. The Allergic Rhinitis Clinical Practice Guidelines of Korean Medicine highlight both herbal medicine and acupuncture as effective and safe treatment modalities for managing AR and preventing recurrence [11].

Despite these promising findings, a critical research gap remains in understanding the factors influencing the utilization and effectiveness of KM combination therapies for pediatric AR patients. Most existing studies have focused on general AR populations, examining treatment patterns, healthcare utilization, or the efficacy of individual KM interventions [6,16]. Furthermore, the Korean healthcare system is uniquely characterized by a dual structure consisting of conventional medicine and KM. In this context, treatment choices—whether conventional, traditional, or integrated—are heavily influenced by patient preference. Therefore, a comprehensive analysis of the factors that guide pediatric patients and their caregivers in choosing combination therapies becomes crucial for developing patient-centered healthcare strategies tailored specifically to this vulnerable group.

This study aims to analyze the factors associated with the utilization of combination therapy involving traditional KM and conventional medicine in pediatric patients with AR. Utilizing large-scale, nationally representative health insurance data from the Healthcare Big Data Hub of the Health Insurance Review and Assessment Service (HIRA), the study investigates the clinical and demographic characteristics of pediatric patients receiving combined KM treatment compared to those receiving conventional medicine alone. Based on the findings, this study is expected to contribute to the development of tailored healthcare policies and strategies that identify the conditions under which the integration of KM into pediatric AR treatment can optimize therapeutic outcomes.

## 2. Materials and Methods

### 2.1. Data Source and Study Population

This study utilized customized research data (No. M20230203001) obtained from the Healthcare Big Data Hub platform of the HIRA. The data were provided following a review by HIRA’s Public Data Provision Deliberation Committee. HIRA’s research data encompass treatment information for the entire population of South Korea, offering extracted, summarized, and anonymized health insurance claim data that have been collected, maintained, and managed to support academic research. The ethical approval of this study was reviewed and exempted by the Institutional Review Board of the Korea Institute of Oriental Medicine (I-2301/001-003, approval date: 13 January 2023). Patient consent was not required due to the retrospective use of anonymous clinical data. The study utilized research data from the HIRA, which is a secondary dataset derived from insurance claims data. The available demographic information includes the patient’s birth month and year, sex, and health insurance eligibility status. Additionally, access to the dataset is time-limited, with a maximum data availability period of two years.

For this study, the dataset included all pediatric patients who received a primary diagnosis of AR between 1 January 2018 and 31 December 2021. Medical records containing information on AR diagnoses were extracted based on diagnosis codes registered by the treating physician. The study population comprised individuals diagnosed with AR during the observation period who were also diagnosed with at least one of the following conditions: atopic dermatitis, asthma, or allergic conjunctivitis, or had an allergy test prescription. Among them, subjects under the age of 18 who were prescribed antihistamines were included in the analysis (Figure 1).

Pediatric allergic rhinitis (AR) patients under 18 years old were extracted from Korea’s national health insurance claims data from 2018 to 2021. After applying exclusion criteria for diseases and treatments known to influence outcomes, patients were classified into those who received KM combination therapy and those who received conventional therapy.

### 2.2. Inclusion and Exclusion Criteria

The inclusion criteria for the study subjects were as follows: (1) patients with a primary diagnosis of AR based on medical benefit claims; (2) patients under 18 years of age at the time of diagnosis; (3) patients diagnosed with at least one of the following conditions—atopic dermatitis, asthma, or allergic conjunctivitis—or prescribed an allergy test within one year before or after the initial AR diagnosis; and (4) patients prescribed second-generation antihistamines. The list and codes for immunological tests, along with the main active ingredient names of second-generation antihistamines, are provided in Appendix A.

The exclusion criteria for the subjects were as follows: (1) patients with secondary diagnoses of cancer, intracranial hemorrhage, cerebral infarction, stroke, other cerebrovascular diseases, or renal failure; and (2) patients who visited medical institutions once or not at all during the observation period. To identify pediatric patients with AR, the Korean Standard Classification of Diseases Version 7 (KCD-7) was used. The KCD-7 is based on the International Classification of Diseases Version 10 (ICD-10) and reflects the unique disease characteristics of Korea. Patients were classified as having AR if their primary diagnosis code included any of the following: J30, J30.1, J30.2, J30.3, or J30.4. The list of diseases and diagnostic codes used for subject extraction in this study can be found in Appendix A.

### 2.3. Use of KM and Covariables

Patients who visited hospitals with a primary diagnosis of AR were classified as receiving traditional KM treatment if the medical institution was designated as a KM hospital or clinic, regardless of whether the care was inpatient or outpatient. To clearly define traditional KM combination therapy, we applied a one-year washout period (January to December 2017) and included only pediatric patients newly diagnosed with AR from 2018 onward. Combination therapy was defined as cases where patients had a cross-visit between KM and conventional medicine (WM) within 10 days of the initial diagnosis. Additionally, treatment continuity was ensured by including only patients who had a follow-up visit within 90 days of their previous visit. To confirm the receipt of KM treatment, only patients who had at least two separate KM visits for the same diagnosis during the observation period were included in the study. To identify the characteristics of pediatric AR patients receiving traditional KM treatment, the following variables were extracted: age, gender, comorbidities and underlying conditions, the timing of onset before and after Coronavirus Disease 2019 (COVID-19), and total number of days that antihistamines were prescribed. Age was categorized as preschool (five years and younger) and school-age (six years and older), while comorbidities included common colds, rhinitis, sinusitis, atopic dermatitis, asthma, pneumonia, and allergic conjunctivitis. Additionally, the onset of AR was categorized as “before COVID” if it occurred between 2018 and 2019 and as “COVID-19 pandemic” if it occurred between 2020 and 2021.

### 2.4. Data Preprocessing and Statistical Analysis

Data analysis was conducted using a remote analysis system provided by the Healthcare Big Data Hub, which was accessible exclusively from pre-authorized, researcher-approved personal computers (PCs). HIRA provides a data analysis environment through a virtualization server that is based on a preapproved media access control and network address and can be assessed using the SAS Enterprise Guide (SAS EG) analytical tool. Therefore, basic data preprocessing was conducted using SAS EG. For data preprocessing, we first extracted individuals under the age of 18 with a primary diagnosis of AR based on their diagnostic codes (J30, J30.1, J30.2, J30.3, and J30.4). We then reviewed the complete diagnostic records of the subjects and excluded those who met the exclusion criteria. Finally, to select the study population, we examined hospital visit records for one year prior to the initial diagnosis within the study period and included only those who had been diagnosed with atopic dermatitis, asthma, or allergic conjunctivitis, or had undergone allergy testing. To assess treatment adherence, we evaluated whether patients had any visits with AR as the primary diagnosis during the one-year follow-up period after treatment completion. Sex and age were included as covariates, with values obtained at the initial visit based on treatment claims data. The presence of comorbidities and underlying conditions was determined according to the diagnostic codes listed in Appendix A. Among the prescribed treatments, only second-generation antihistamines were included in the study. To compare the characteristics of patients receiving combined traditional KM and conventional medicine treatment versus those receiving conventional medicine alone, Χ2 tests were employed for categorical variables. For continuous variables, crosstabs were used. Statistical significance was determined at a threshold of *p* < 0.05. Descriptive statistics were used to calculate yearly prevalence rates. Logistic regression analysis was performed to identify factors associated with combined traditional KM treatment. All statistical analyses were conducted using the Statistical Analysis System (SAS) Enterprise Guide V.9.4.2 (SAS Institute, Cary, NC, USA) installed on a virtualized PC.

## 3. Results

### 3.1. Baseline Characteristics of Participants

Over the past four years, a total of 696,182 pediatric patients with AR were included in this study, among whom 61,745 (8.9%) received concurrent traditional KM treatment. Regarding gender distribution of the entire cohort, males accounted for 55.9%, with a statistically significant predominance of male patients in the traditional KM treatment group (*p* < 0.0001). Age-group analysis revealed a statistically significant older average age in the traditional KM treatment group (*p* = 0.006), with a higher proportion of school-age children (*p* < 0.0001).

The prevalence of comorbidities, including common colds, rhinitis, atopic dermatitis, asthma, and allergic conjunctivitis, was also higher in the traditional KM treatment group (*p* < 0.0001). Furthermore, both the total treatment duration and the number of days antihistamines were prescribed were higher in the traditional KM treatment group. When considering the timing of AR onset relative to the COVID-19 pandemic, the traditional KM treatment group had a higher prevalence of diagnoses before the onset of COVID-19 compared to the control group (Table 1).

### 3.2. Factors Associated with the Use of KM

The results of a multivariate logistic regression analysis examining factors associated with concurrent traditional KM treatment in pediatric patients with AR are presented in Table 2. The likelihood of receiving traditional KM treatment was lower for preschool-age children (OR: 0.40, 95% CI: 0.23 to 0.69), while it was significantly higher for patients diagnosed during colder seasons (OR: 1.62, 95% CI: 1.11 to 2.38). Additionally, patients with chronic rhinitis/nasopharyngitis (OR: 1.94, 95% CI: 1.31 to 2.86) or atopic dermatitis (OR: 1.62, 95% CI: 1.10 to 2.39) were more likely to receive KM treatment (Figure 2).

## 4. Discussion

This study investigated the factors associated with the use of combination therapy involving traditional KM and conventional medicine in pediatric patients with AR using data from the Healthcare Big Data Hub of HIRA. The findings provided critical insights into the characteristics and factors influencing the use of combination therapy, with essential implications for clinical practice, healthcare policy, and future research.

We focused our analysis on pediatric patients with AR due to its high prevalence and the associated medical costs in this age group. Data from the Korean National Health and Nutrition Examination Survey, a nationwide cross-sectional study, estimated the prevalence of AR at 9.0% in infants, 20.2% in preschool children, 27.6% in school-age children, 17.1% in adults, and 6.9% in the elderly [17]. Furthermore, the mechanisms, medical history, and treatment outcomes of AR differ significantly between adults and children [18]. These findings underscore the importance of AR as a major health concern in pediatric populations.

Given these statistics, focusing on pediatric patients was crucial to gain tailored insights into their unique characteristics, treatment patterns, and healthcare needs. Understanding the factors influencing treatment choices and outcomes in this vulnerable group can help optimize care strategies and inform the development of targeted healthcare policies.

The results showed that 8.9% of pediatric AR patients received combination therapy with KM and conventional medicine. School-aged children were more likely to receive combination therapy than preschool-aged children, potentially reflecting a preference for complementary approaches to manage chronic or refractory symptoms in older patients. Comorbid conditions such as rhinitis and atopic dermatitis were strongly associated with the use of combination therapy, highlighting the role of KM in addressing complex clinical profiles with multiple allergic conditions.

These findings align with previous studies demonstrating the efficacy of KM, including herbal medicines and acupuncture, in managing allergic symptoms and improving quality of life [7,14,19]. While prior research has predominantly focused on general AR populations, this study emphasizes the unique characteristics and treatment preferences of pediatric patients, which are crucial for tailoring interventions to specific demographic groups.

The association of combination therapy with older age groups and comorbid conditions underscores the importance of integrating KM into pediatric AR care. KM offers a complementary approach to symptom management and quality-of-life improvement, particularly for patients with persistent or multifaceted symptoms. Developing integrative care models that combine the strengths of KM and conventional medicine could enhance treatment outcomes and patient satisfaction. Furthermore, such integration could be extended to other chronic conditions, such as common colds and atopic dermatitis in children, where multi-modal treatment approaches may yield similar benefits.

A study comparing the outcomes of conventional medicine and KM treatments for AR using HIRA claims data found that the conventional treatment group had a 1.701 times higher risk of recurrence than the KM treatment group. Additionally, the risk of developing asthma was 1.609 times higher, and the risk of atopic dermatitis was 1.098 times higher in the WM treatment group compared to the KM treatment group [20]. These findings emphasize the potential advantages of incorporating KM into pediatric AR care and suggest broader applications for integrative treatment models in managing AR.

The analysis also revealed a decrease in KM utilization during the COVID-19 pandemic, which may be attributed to reduced access to traditional KM clinics or shifts in healthcare-seeking behaviors during the public health crisis. A recent study reported that the number of visits to KM clinics remained stable at approximately 100 million annually until 2019, after which visits sharply declined by 10.30% in 2020 and continued to decrease thereafter [21]. These findings underscore the necessity of ensuring healthcare access during emergencies, particularly for complementary and alternative medicine users.

During the COVID-19 pandemic in South Korea, outdoor activities were restricted to prevent the spread of infection, which led individuals to prioritize essential conventional medical treatments. Meanwhile, optional treatments, such as KM, were likely avoided. This behavior reflects the impact of public health measures and risk aversion on treatment choices during the pandemic.

The observed decline in KM utilization during the COVID-19 pandemic highlights the need to address barriers to healthcare access during public health emergencies. Telemedicine and digital health solutions could be explored as viable alternatives to ensure the continuity of KM care in similar future scenarios. Additionally, the role of KM in addressing post-viral symptoms or pandemic-related stress disorders presents a promising area for future research and application.

The study also found that combination therapy was associated with longer treatment durations and higher out-of-pocket expenses compared to conventional medicine alone. While these findings suggest increased healthcare utilization, they also raise important questions regarding the long-term cost-effectiveness of KM combination therapy. Future studies should evaluate whether the upfront costs associated with KM lead to downstream savings through reduced complications, fewer hospitalizations, or improved patient outcomes.

Despite these valuable insights, several limitations should be acknowledged. First, the study design initially aimed to compare three groups, namely, KM-only treatment, conventional medicine-only treatment, and combination therapy. However, the KM-only group was too small to allow for meaningful analysis, reflecting the relatively low proportion of patients in Korea who opt for KM-only treatment. Future studies assessing the efficacy and safety of KM as a standalone treatment may require clinical trials or alternative data collection strategies to ensure adequate representation of KM-only patients. Second, the analysis was limited to reimbursed treatments, excluding non-covered KM health insurance services such as herbal decoctions. This omission may underestimate the actual use of KM and its associated benefits. Future research should include non-reimbursed treatments to provide a more comprehensive assessment of KM utilization. Third, the reliance on administrative claims data precluded the analysis of clinical variables, such as disease severity, symptom burden, and patient preferences. Incorporating these factors in future research would enhance the understanding of treatment selection and outcomes. Longitudinal studies are also warranted to assess the long-term effectiveness and cost-efficiency of KM combination therapy. Fourth, the observational design limits the ability to infer causality. While significant associations were identified, randomized controlled trials are needed to establish causal effects of KM combination therapy on pediatric AR management. Future research could also explore the scalability of integrative KM models to other healthcare settings and populations. Finally, this study did not include correlational analysis to examine the relationships between factors associated with combination therapy and demographic characteristics. Our methodological approach primarily focused on between-group differences, rather than exploring the associations between demographic variables and treatment factors within each treatment group.

Despite these limitations, our study presents several key strengths and significant implications. Foremost, this research analyzed nationwide health insurance claims data encompassing the entire Korean population, providing a comprehensive large-population perspective on pediatric AR treatment patterns. This approach offers robust real-world evidence that may not be captured in smaller clinical trials or regional studies. The nationwide scope enhances the generalizability of our findings across different geographic regions and healthcare settings within Korea.

Our findings regarding the potential benefits of combination therapy have important clinical implications. The observed differences in healthcare utilization patterns between treatment groups suggest that integrating KM with conventional medicine may offer complementary benefits for pediatric AR management. For clinicians, these results provide preliminary evidence to consider integrated approaches for patients with suboptimal responses to conventional treatments alone. However, we acknowledge that treatment decisions should be individualized based on patient preferences, clinical presentation, and available resources.

From a policy perspective, our findings highlight the need for healthcare systems to recognize and potentially support integrative medicine approaches for pediatric AR. Current reimbursement policies may influence treatment selection and access to complementary therapies. Policymakers could consider expanding coverage for evidence-based KM interventions to improve accessibility and reduce financial barriers for families seeking integrative care options.

Future research directions should address the limitations identified in this study. Longitudinal studies are needed to evaluate the long-term effectiveness, safety, and cost-efficiency of KM combination therapy for pediatric AR. Randomized controlled trials comparing conventional medicine, KM, and combination approaches would provide stronger causal evidence regarding treatment efficacy. Additionally, qualitative research exploring patient experiences, preferences, and decision-making processes would enhance our understanding of treatment selection factors. Future studies should also incorporate clinical variables, such as symptom severity, quality of life measures, and treatment adherence, to provide a more comprehensive assessment of outcomes beyond healthcare utilization metrics.

## 5. Conclusions

This study highlights the factors influencing the use of KM combination therapy in pediatric AR patients, offering valuable insights for clinical practice and policy. The findings emphasize the need for integrative care models, data-driven policy adjustments, and further research to optimize the role of KM in managing pediatric AR.

Despite methodological constraints, our large-population analysis provides significant real-world evidence on treatment patterns in Korea. The observed potential benefits of combining conventional medicine with KM warrant further investigation through more rigorous research designs across diverse populations. By addressing existing gaps in knowledge and practice, this study contributes to advancing patient-centered, evidence-based approaches in pediatric healthcare.

## Figures and Tables

**Figure 1 healthcare-13-00875-f001:**
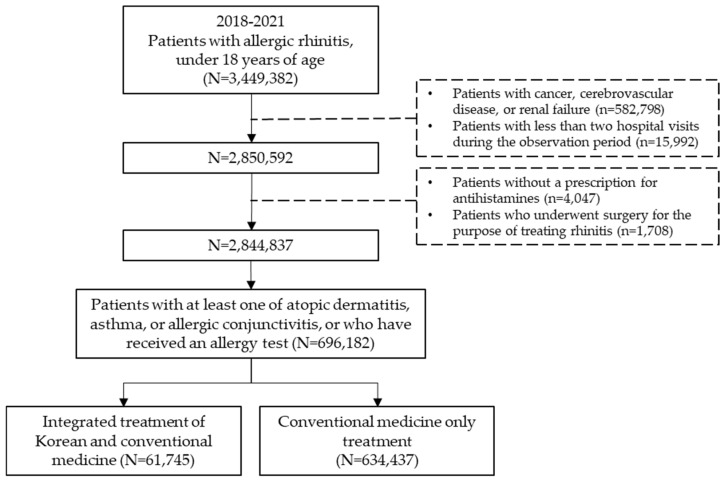
Schematic flow chart.

**Figure 2 healthcare-13-00875-f002:**
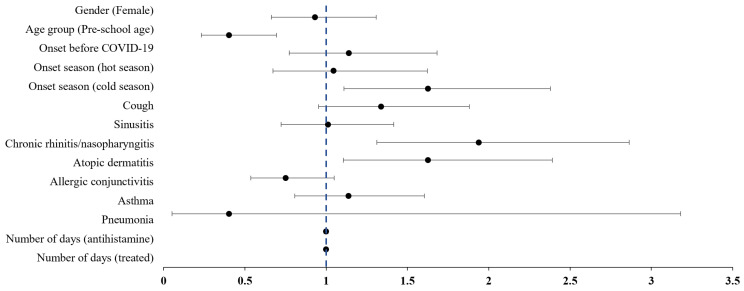
Forest plot of the adjusted odds ratios on the combination treatment with Korean Medicine in pediatric allergic rhinitis. Bullets represent odds ratios, and the area of the horizontal line represents confidence intervals.

**Table 1 healthcare-13-00875-t001:** Baseline characteristics of participants.

	Total n (%) or Mean ± SD	Conventional Medicine Only n (%) or Mean ± SD	Korean–Conventional Medicine n (%) or Mean ± SD	*p*-Value
Patients	696,182	634,437 (91.1)	61,745 (8.9)	
Age	7.0 ± 4.8	6.9 ± 4.8	8.1 ± 5.1	<0.0001
Age group				
Infants (<2)	97,903 (14.1)	92,480 (14.5)	5423 (8.8)	0.006
Preschool children (2–5)	217,394 (31.2)	199,510 (31.5)	17,884 (29.0)	
School-age children (6–18)	380,885 (54.7)	342,447 (54.0)	38,438 (62.2)	
Sex				
Male	389,312 (55.9)	352,385 (55.5)	36,927 (59.8)	<0.0001
Female	306,870 (44.1)	285,052 (44.5)	24,818 (40.2)	
Health insurance subscriber classification				
Workplace subscribers	684,659 (98.3)	624,111 (98.4)	60,548 (89.1)	<0.0001
Regional subscribers	11,523 (1.7)	10,326 (1.6)	1197 (1.9)	
Comorbidity				
Acute upper respiratory tract infection	395,017 (56.7)	355,597 (56.1)	39,420 (63.8)	<0.0001
Chronic rhinitis/nasopharyngitis	66,920 (9.6)	55,287 (8.7)	11,633 (18.8)	<0.0001
Sinusitis	375,220 (53.9)	341,729 (53.9)	33,491 (54.2)	0.073
Atopic dermatitis	197,413 (28.4)	179,191 (28.2)	18,232 (29.5)	<0.0001
Asthma	384,946 (55.3)	350,165 (55.2)	34,781 (56.3)	<0.0001
Pneumonia	6013 (0.9)	5474 (0.9)	539 (0.9)	0.785
Allergic conjunctivitis	366,027 (52.6)	330,376 (52.1)	35,651 (57.7)	<0.0001
Days of treatment	306.7 ± 301.7	297.6 ± 295.8	399.3 ± 341.9	<0.0001
Treatment within 1 year				
Cured	199,931 (28.7)	182,547 (28.8)	17,384 (28.2)	<0.0001
Recurred	496,251 (71.3)	451,890 (71.2)	44,361 (71.8)	
Medical expenses				
Total medical expenses	249,065 ± 1,238,649	237,351 ± 1,282,702	369,434 ± 614,061	<0.0001
Out-of-pocket medical expenses	57,748 ± 151,558	54,582 ± 151,868	90,282 ± 144,371	<0.0001
Onset period I				
Cold season	233,689 (33.6)	211,957 (33.4)	21,732 (35.2)	
Hot season	180,491 (25.9)	164,662 (26.0)	15,829 (25.6)	0.011
Transitional season	282,002 (40.5)	257,818 (40.6)	24,184 (39.2)	
Onset period II				
Before COVID-19 (2018–2019)	336,707 (48.4)	300,667 (47.4)	26,040 (58.4)	<0.0001
COVID-19 pandemic (2020–2021)	359,475 (51.6)	333,770 (52.6)	25,705 (41.6)	
Antihistamine prescription days	8.1 ± 14.9	7.9 ± 14.2	8.9 ± 18.5	0.407

COVID-19: Coronavirus Disease 2019, SD: standard deviations.

**Table 2 healthcare-13-00875-t002:** Multivariate logistic regression analysis on the combination therapy with Korean medicine in pediatric allergic rhinitis.

		Estimate (SE)	Odds Ratio	95% CI	*p*-Value
Intercept		−1.88 (0.38)	–	–	<0.0001
Gender	Male	–	–	–	–
	Female	−0.07 (0.17)	0.93	0.66–1.31	0.6784
Age group	School age	–	–	–	–
	Preschool age	−0.91 (0.28)	0.40	0.23–0.69	0.0011
Onset before COVID-19	No	–	–	–	–
	Yes	0.13 (0.20)	1.14	0.77–1.68	0.5107
Onset season	Transitional	–	–	–	–
	Hot	0.04 (0.22)	1.04	0.67–1.62	0.8449
	Cold	0.48 (0.19)	1.62	1.11–2.38	0.0125
Cough	No	–	–	–	–
	Yes	0.29 (0.17)	1.34	0.95–1.88	0.0930
Sinusitis	No	–	–	–	–
	Yes	0.01 (0.17)	1.01	0.72–1.42	0.9445
Chronic rhinitis/ nasopharyngitis	No	–	–	–	–
	Yes	0.66 (0.20)	1.94	1.31–2.86	0.0009
Atopic dermatitis	No	–	–	–	–
	Yes	0.48 (0.20)	1.62	1.10–2.39	0.0136
Allergic conjunctivitis	No	–	–	–	–
	Yes	−0.29 (0.17)	0.75	0.54–1.05	0.0940
Asthma	No	–	–	–	–
	Yes	0.13 (0.17)	1.14	0.81–1.60	0.4643
Pneumonia	No	–	–	–	–
	Yes	−0.09 (1.05)	0.40	0.05–3.18	0.3884
Number of days (antihistamine)		−0.00 (0.00)	0.99	0.99–1.01	0.8913
Number of days (treated)		0.00 (0.00)	1.000	1.00–1.00	0.1826

Adjusted odds ratios are presented as the results of multivariate logistic regression.

## Data Availability

Restrictions apply to the availability of these data. Data were obtained from the Healthcare Big Data Hub of Health Insurance Review and Assessment (HIRA) in Korea and are available at https://opendata.hira.or.kr (accessed on 7 June 2024) with the permission of HIRA.

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
