# Peer review of "Factors Associated with Combination Therapy Involving Traditional Korean Medicine in Pediatric Allergic Rhinitis Patients: A Retrospective Cohort Study Using National Health Insurance Data"

_healthcare, 2025, doi:10.3390/healthcare13080875_

Round 1

Reviewer 1 Report

Comments and Suggestions for Authors

Dear Editor, 

Thank you for the opportunity to review this manuscript. Below, I outline my recommendations for improvement.

Introduction:

The introduction lacks a comprehensive overview of the study participants. Providing details about the demographic characteristics of the participants would enhance clarity.

The research problem is not clearly defined. The authors should explicitly state the gap in the literature that this study aims to address.

The significance of the study needs to be articulated. Explaining the potential impact of the findings on clinical practice and public health would strengthen this section.

Methods:

Ethical Considerations: The manuscript does not mention any ethical approval or informed consent procedures. It is crucial to include details about the ethical guidelines followed, including IRB approval and data confidentiality measures.

Data Collection Procedure: The methodology lacks a detailed explanation of how data was collected. Providing a step-by-step description of data extraction from the National Health Insurance database would enhance transparency and reproducibility.

Statistical Analysis:

The current statistical methods do not include correlational analysis. It is recommended to incorporate correlation tests to examine the relationships between the factors associated with combination therapy and sample demographic characteristics.

A correlation table summarizing these associations should be included to provide a clearer understanding of the interplay between variables.

Discussion:

While comprehensive, the discussion section requires further elaboration on the justifications for the study’s results. The authors should provide stronger theoretical or empirical support for their findings.

The study’s strengths and limitations should be explicitly discussed. Identifying potential biases, data constraints, and generalizability concerns will improve the manuscript’s transparency.

Future implications of the findings should be addressed, including recommendations for clinical practice and policy development.

The authors should suggest directions for future research, such as longitudinal studies or randomized controlled trials to validate their findings.

Regards, 

Author Response

Introduction:

Comments 1: The introduction lacks a comprehensive overview of the study participants. Providing details about the demographic characteristics of the participants would enhance clarity.

Response 1: We have supplemented the "2.1. Data Source and Study Population" section with additional explanations regarding the demographic characteristics of study participants and the overall dataset.

Line 101-107: “The study utilized research data from the HIRA, which is a secondary dataset derived from insurance claims data. The available demographic information includes the patient’s birth month and year, sex, and health insurance eligibility status. Additionally, access to the dataset is time-limited, with a maximum data availability period of two years. For this study, the dataset included all pediatric patients who received a primary diagnosis of allergic rhinitis (AR) between January 1, 2018, and December 31, 2021.”

Comments 2: The research problem is not clearly defined. The authors should explicitly state the gap in the literature that this study aims to address.

Response 2: We appreciate your thoughtful feedback regarding the clarity of our research problem. While the research gap was initially addressed in the introduction (lines 59-65), we acknowledge the need for a more explicit and comprehensive articulation of the literature gap.

Line 66-76: “Despite these promising findings, a critical research gap remains in understanding the factors influencing the utilization and effectiveness of KM combination therapies for pediatric AR patients. Most existing studies have focused on general AR populations, examining treatment patterns, healthcare utilization, or the efficacy of individual KM interventions [6,16]. Furthermore, the Korean healthcare system is uniquely characterized by a dual structure consisting of conventional medicine and Korean traditional medicine. In this context, treatment choices—whether conventional, traditional, or integrated—are heavily influenced by patient preference. Therefore, a comprehensive analysis of the factors that guide pediatric patients and their caregivers in choosing combination therapies becomes crucial for developing patient-centered healthcare strategies tailored specifically to this vulnerable group.”

Comments 3: The significance of the study needs to be articulated. Explaining the potential impact of the findings on clinical practice and public health would strengthen this section.

Response 3: To clarify the significance of the study, we have additionally provided the following explanation.

Line 83-85: “Based on the findings, this study is expected to contribute to the development of tailored healthcare policies and strategies that identify the conditions under which the integration of KM into pediatric AR treatment can optimize therapeutic outcomes.”

Methods:

Comments 4: Ethical Considerations: The manuscript does not mention any ethical approval or informed consent procedures. It is crucial to include details about the ethical guidelines followed, including IRB approval and data confidentiality measures.

Response 4: The relevant content was originally stated in the final Acknowledgement section of our manuscript. However, in response to the review comments, we have incorporated it into the "2.1. Data Source and Study Population" section as follows.

Line 94-96: “The ethical approval of this study was reviewed and exempted by the Institutional Re-view Board of the Korea Institute of Oriental Medicine (I-2301/001-003, Approval Date: January 13, 2023). Patient consent was not required due to the retrospective use of anonymous clinical data.”

Comments 5: Data Collection Procedure: The methodology lacks a detailed explanation of how data was collected. Providing a step-by-step description of data extraction from the National Health Insurance database would enhance transparency and reproducibility.

Response 5: In accordance with your feedback, we have added a step-by-step explanation of the data extraction process from the National Health Insurance database in the "2.4. Data Preprocessing and Statistical Analysis" section.

Line 155-171: “HIRA provides a data analysis environment through a virtualization server that is based on a preapproved media access control and network address and can be assessed with the SAS Enterprise Guide (SAS EG) analytical tool. Therefore, basic data preprocessing was conducted using SAS EG. For data preprocessing, we first extracted individuals under the age of 18 with a primary diagnosis of allergic rhinitis (AR) based on the diagnostic codes (J30, J30.1, J30.2, J30.3, J30.4). We then reviewed the complete diagnostic records of the subjects and excluded those who met the exclusion criteria. Finally, to select the study population, we examined hospital visit records for one year prior to the initial diagnosis within the study period and included only those who had been diagnosed with atopic dermatitis, asthma, or allergic conjunctivitis, or had undergone allergy testing. To assess treatment adherence, we evaluated whether patients had any visits with AR as the primary diagnosis during the one-year follow-up period after treatment completion. Sex and age were included as covariates, with values obtained at the initial visit based on treatment claims data. The presence of comorbidities and underlying conditions was determined according to the diagnostic codes listed in Supplementary Table 2. Among the prescribed treatments, only second-generation antihistamines were included in the study.”

Statistical Analysis:

Comments 6: The current statistical methods do not include correlational analysis. It is recommended to incorporate correlation tests to examine the relationships between the factors associated with combination therapy and sample demographic characteristics.

A correlation table summarizing these associations should be included to provide a clearer understanding of the interplay between variables.

Response 6: Thank you for your valuable suggestion. However, our study did not include correlational analysis as part of the statistical methodology. Additionally, due to data access restrictions, it is currently not possible to perform further analyses. We acknowledge the importance of understanding the relationships between factors associated with combination therapy and demographic characteristics, and we will consider incorporating correlation analyses in future studies when access to relevant data is available. Additionally, to address the limitation of not conducting a correlation analysis, we have included the following statement in the Discussion section.

Line 305-309: “Finally, this study did not include correlational analysis to examine the relationships between factors associated with combination therapy and demographic characteristics. Our methodological approach primarily focused on between-group differences, rather than exploring the associations between demographic variables and treatment factors within each treatment group.”

Discussion:

Comments 7: While comprehensive, the discussion section requires further elaboration on the justifications for the study’s results. The authors should provide stronger theoretical or empirical support for their findings. The study’s strengths and limitations should be explicitly discussed. Identifying potential biases, data constraints, and generalizability concerns will improve the manuscript’s transparency.

Response 7: We sincerely appreciate the reviewer's insightful feedback. In light of the comment, we have substantially enhanced the discussion section to incorporate and address the suggested points, thereby strengthening the overall scholarly contribution of our manuscript.

Line 310-316: “Despite these limitations, our study presents several key strengths and significant implications. Foremost, this research analyzed nationwide health insurance claims data encompassing the entire Korean population, providing a comprehensive large-population perspective on pediatric AR treatment patterns. This approach offers robust real-world evidence that may not be captured in smaller clinical trials or regional studies. The nationwide scope enhances the generalizability of our findings across different geographic regions and healthcare settings within Korea.”

Comments 8: Future implications of the findings should be addressed, including recommendations for clinical practice and policy development.

Response 8: We have revised the discussion section in accordance with the reviewer's comment, elaborating on the future implications of our findings with specific recommendations for clinical practice and policy development.

Line 317-330: Our findings regarding the potential benefits of combination therapy have important clinical implications. The observed differences in healthcare utilization patterns between treatment groups suggest that integrating KM with conventional medicine may offer complementary benefits for pediatric AR management. For clinicians, these results provide preliminary evidence to consider integrated approaches for patients with suboptimal responses to conventional treatments alone. However, we acknowledge that treatment decisions should be individualized based on patient preferences, clinical presentation, and available resources.

From a policy perspective, our findings highlight the need for healthcare systems to recognize and potentially support integrative medicine approaches for pediatric AR. Current reimbursement policies may influence treatment selection and access to complementary therapies. Policymakers could consider expanding coverage for evidence-based KM interventions to improve accessibility and reduce financial barriers for families seeking integrative care options.

Comments 9: The authors should suggest directions for future research, such as longitudinal studies or randomized controlled trials to validate their findings.

Response 9: We have revised the discussion section in accordance with the reviewer's comment.

Line 331-340: Future research directions should address the limitations identified in this study. Longitudinal studies are needed to evaluate the long-term effectiveness, safety, and cost-efficiency of KM combination therapy for pediatric AR. Randomized controlled trials comparing conventional medicine, KM, and combination approaches would provide stronger causal evidence regarding treatment efficacy. Additionally, qualitative research exploring patient experiences, preferences, and decision-making processes would enhance our understanding of treatment selection factors. Future studies should also incorporate clinical variables, such as symptom severity, quality of life measures, and treatment adherence, to provide a more comprehensive assessment of outcomes beyond healthcare utilization metrics.

Reviewer 2 Report

Comments and Suggestions for Authors

This manuscript presents an important and well-structured study analyzing the factors influencing the utilization of combination therapy involving traditional Korean medicine (KM) and conventional medicine in pediatric allergic rhinitis (AR) patients. The study is based on a large-scale dataset from the National Health Insurance Service in South Korea, providing valuable insights into treatment patterns and healthcare utilization in pediatric AR.

The research is relevant to pediatric allergy and complementary medicine, and its findings have potential implications for clinical practice and healthcare policy. 

Suggestions for improvements:

  1. Abstract
  2. Introduction. While it provides a good background on allergic rhinitis and the role of KM, it lacks a clear rationale for investigating combination therapy specifically. A brief discussion of why patients and clinicians may choose combination therapy over conventional or KM-alone treatment would strengthen the study’s foundation.
  3. Methods. The inclusion and exclusion criteria are generally well-defined, but some additional clarifications are needed. Why were only patients prescribed second-generation antihistamines included? 
  4. It is not entirely clear how KM and conventional medicine were classified as "combination therapy." Provide a clear operational definition of combination therapy—was it based on concurrent use, sequential use, or a minimum duration threshold?
  5. The study categorizes onset seasons as cold, hot, and transitional. However, the rationale for this classification is not discussed. Reference climatological data or prior studies to support the chosen classification.
  6. Results. Lines 155–168 completely duplicate the text from lines 141–153. Lines 141–153 should be removed.
  7. The greatest concern is the inclusion of acute conditions, including acute rhinitis, as comorbidities. It is possible that in this case, traditional Korean medicine was used to treat an acute respiratory infection with rhinitis symptoms rather than allergic rhinitis itself. It sounds somewhat unusual to classify rhinitis as a comorbidity of allergic rhinitis. This concept needs to be reconsidered. The same pattern was observed for the cold season, which may also be related to the increase in respiratory infections during this time and the use of Korean medicine for their treatment. It would be reasonable to exclude children with acute conditions (acute respiratory infections, including acute rhinitis) from the study to obtain reliable data specifically on the treatment of allergic rhinitis using various methods.
  8. The authors note a decline in KM utilization during the COVID-19 pandemic but does not explore the possible reasons behind this trend in detail. Discuss whether clinic closures, telemedicine barriers, or patient risk aversion influenced this decline.
  9. Discussion. Provide more evidence of how KM interventions (e.g., acupuncture, herbal medicine) might physiologically modulate AR symptoms.
  10. Expand the discussion on how this study aligns with or contradicts previous research on KM use in AR.
  11. The manuscript suggests that findings could guide healthcare policies, but it does not specify actionable recommendations.  Discuss potential policy changes, such as insurance coverage expansion for KM or integrated care models.
  12. The study appropriately acknowledges some limitations, but additional considerations should be discussed. While the study identifies associations, it cannot establish causality. Highlight the need for prospective or interventional studies to confirm findings.
  13. The tables and figures are well-structured and provide comprehensive statistical details.
  14. The conclusion should be more specific, taking into account the comments provided above.

Author Response

Suggestions for improvements:

Comments 1: Introduction. While it provides a good background on allergic rhinitis and the role of KM, it lacks a clear rationale for investigating combination therapy specifically. A brief discussion of why patients and clinicians may choose combination therapy over conventional or KM-alone treatment would strengthen the study’s foundation.

Response 1: We have revised the introduction section in accordance with the reviewer's comment.

Line 70-76: “Furthermore, the Korean healthcare system is uniquely characterized by a dual structure consisting of conventional medicine and Korean traditional medicine. In this context, treatment choices—whether conventional, traditional, or integrated—are heavily influenced by patient preference. Therefore, a comprehensive analysis of the factors that guide pediatric patients and their caregivers in choosing combination therapies becomes crucial for developing patient-centered healthcare strategies tailored specifically to this vulnerable group.”

Comments 2: Methods. The inclusion and exclusion criteria are generally well-defined, but some additional clarifications are needed. Why were only patients prescribed second-generation antihistamines included? 

Response 2: With the recent development of second-generation antihistamines that significantly reduce drowsiness, first-generation antihistamines are rarely prescribed long-term for patients with allergic rhinitis. Currently, first-generation antihistamines are primarily used in combination cold medications, as sleep aids, or as antiemetic drugs to suppress vomiting. Thus, we exclusively included patients prescribed second-generation antihistamines

Comments 3: It is not entirely clear how KM and conventional medicine were classified as "combination therapy." Provide a clear operational definition of combination therapy—was it based on concurrent use, sequential use, or a minimum duration threshold?

Response 3: Thank you for your insightful comment. In our study, combination therapy was defined as cases in which patients received at least two or more KM treatments in addition to conventional medicine during the treatment period. To clarify this definition, we have revised the manuscript to explicitly state the criteria used for classifying combination therapy.

Line 141-143: “To clearly define traditional KM combination therapy, only patients who received at least two separate KM treatment sessions during the observation period were included in the study.”

Comments 4: The study categorizes onset seasons as cold, hot, and transitional. However, the rationale for this classification is not discussed. Reference climatological data or prior studies to support the chosen classification.

Response 4: South Korea has four distinct seasons, and based on data from its capital, Seoul, the average temperature in the hottest month (August) is 29.3°C, while the coldest month (January) has an average temperature of -0.5°C. Considering this seasonal variation, we classified seasons based on monthly average temperatures to reflect seasonality in our study design. In this study, the cold season was defined as November to February, the hot season as June to September, and the remaining months were categorized as the transitional season.

Comments 5: Results. Lines 155–168 completely duplicate the text from lines 141–153. Lines 141–153 should be removed.

Response 5: Thank you for your careful review. We have removed the duplicated section in lines 141–153 as suggested.

Comments 6: The greatest concern is the inclusion of acute conditions, including acute rhinitis, as comorbidities. It is possible that in this case, traditional Korean medicine was used to treat an acute respiratory infection with rhinitis symptoms rather than allergic rhinitis itself. It sounds somewhat unusual to classify rhinitis as a comorbidity of allergic rhinitis. This concept needs to be reconsidered. The same pattern was observed for the cold season, which may also be related to the increase in respiratory infections during this time and the use of Korean medicine for their treatment. It would be reasonable to exclude children with acute conditions (acute respiratory infections, including acute rhinitis) from the study to obtain reliable data specifically on the treatment of allergic rhinitis using various methods.

Response 6: Thank you for raising this important concern. Our study specifically included pediatric patients whose primary diagnosis was allergic rhinitis. Comorbid conditions analyzed in this study—such as common colds, rhinitis, sinusitis, atopic dermatitis, asthma, pneumonia, and allergic conjunctivitis—were identified based on secondary diagnoses frequently observed alongside allergic rhinitis. Thus, these conditions represent common accompanying disorders rather than primary acute conditions that could confound the diagnosis. We appreciate your insightful suggestion and will clarify this aspect in the manuscript to prevent potential confusion.

Comments 7: The authors note a decline in KM utilization during the COVID-19 pandemic but does not explore the possible reasons behind this trend in detail. Discuss whether clinic closures, telemedicine barriers, or patient risk aversion influenced this decline.

Response 7: Thank you for your careful review. We have added another evidence about the KM utilization during COVID-19.

Line 262-265: A recent study reported that the number of visits to Korean medicine clinics remained stable at approximately 100 million annually until 2019, after which visits sharply declined by 10.30% in 2020 and continued to decrease thereafter [19].

Comments 8: Discussion. Provide more evidence of how KM interventions (e.g., acupuncture, herbal medicine) might physiologically modulate AR symptoms.

Response 8: Thank you for your careful review. We have added pathophysiological evidence supporting KM interventions to the introduction section.

Line 54-57: “Certain herbal formulas have been shown to modulate immune responses by reducing serum IgE levels, suppressing inflammatory mediators, and restoring the balance between Th1 and Th2 cell activities, ultimately alleviating allergic inflammation [12].”

Line 60-63: “Acupuncture exerts therapeutic effects by modulating neuroimmune responses, balancing pro-inflammatory and anti-inflammatory cytokines, and reducing allergen-specific IgE levels and inflammatory neuropeptides, thus effectively improving clinical symptoms and quality of life in patients with AR​ [15].”

Comments 9: Expand the discussion on how this study aligns with or contradicts previous research on KM use in AR.

Response 9: Thank you for your insightful comment. As you pointed out, a direct comparison with previous studies is challenging because, to our knowledge, this is the first study analyzing combination therapy involving Korean Medicine (KM) and conventional medicine for pediatric allergic rhinitis (AR) using comprehensive national health insurance data. Although many existing studies have evaluated the effectiveness or utilization of individual KM interventions (e.g., herbal medicines or acupuncture) in AR management, none have specifically examined the factors associated with choosing combined KM and conventional treatments in a nationwide pediatric population. However, our findings align with earlier studies that highlight the potential benefits of KM treatments, such as symptom relief, improved quality of life, and reduced recurrence risk in AR patients. Moreover, our observation of increased KM use among older pediatric patients and those with comorbid allergic conditions (e.g., atopic dermatitis) is consistent with previous literature that suggests KM’s effectiveness for chronic and complex allergic conditions. We have expanded the discussion section to acknowledge this clearly and to highlight the novel contribution of our research in filling the existing literature gap through analysis of national-level data.

Comments 10: The manuscript suggests that findings could guide healthcare policies, but it does not specify actionable recommendations.  Discuss potential policy changes, such as insurance coverage expansion for KM or integrated care models.

Response 10: We have revised the discussion section in accordance with the reviewer's comment.

Line 325-330: From a policy perspective, our findings highlight the need for healthcare systems to recognize and potentially support integrative medicine approaches for pediatric AR. Current reimbursement policies may influence treatment selection and access to complementary therapies. Policymakers could consider expanding coverage for evidence-based KM interventions to improve accessibility and reduce financial barriers for families seeking integrative care options.

Comments 11: The study appropriately acknowledges some limitations, but additional considerations should be discussed. While the study identifies associations, it cannot establish causality. Highlight the need for prospective or interventional studies to confirm findings.

Response 11: We have revised the discussion section in accordance with the reviewer's comment.

Line 331-340: Future research directions should address the limitations identified in this study. Longitudinal studies are needed to evaluate the long-term effectiveness, safety, and cost-efficiency of KM combination therapy for pediatric AR. Randomized controlled trials comparing conventional medicine, KM, and combination approaches would provide stronger causal evidence regarding treatment efficacy. Additionally, qualitative research exploring patient experiences, preferences, and decision-making processes would enhance our understanding of treatment selection factors. Future studies should also incorporate clinical variables, such as symptom severity, quality of life measures, and treatment adherence, to provide a more comprehensive assessment of outcomes beyond healthcare utilization metrics.

Comments 12: The tables and figures are well-structured and provide comprehensive statistical details.

Comments 13: The conclusion should be more specific, taking into account the comments provided above.

Response 13: We have revised the conclusion section in accordance with the reviewer's comment.

Line 347-352: Despite methodological constraints, our large-population analysis provides significant real-world evidence on treatment patterns in Korea. The observed potential benefits of combining conventional medicine with KM warrant further investigation through more rigorous research designs across diverse populations. By addressing existing gaps in knowledge and practice, this study contributes to advancing patient-centered, evidence-based approaches in pediatric healthcare.

Reviewer 3 Report

Comments and Suggestions for Authors

The study examines factors influencing the use of combination therapy involving traditional Korean medicine (KM) and conventional medicine in pediatric allergic rhinitis (AR) patients, finding that school-age children, winter season, and comorbidities like rhinitis and atopic dermatitis are key predictors, while KM therapy is associated with longer treatment durations and higher costs.

Title: The current title is unclear; it should explicitly mention “Traditional Korean Medicine…” to accurately reflect the study’s focus.

Figures: The figures need more detailed explanations. Currently, they only have titles without legends or descriptions. Each abbreviation used in a figure must be defined in the legend, even if it has been introduced in the main text.

Supplementary Table: As indicated by its name, the supplementary table should be provided as a separate file.

Supplementary Table 1: The data in this table is unclear. The distinction between “Allergen-Specific Immunoglobulin-IgE (D744001), Allergen-Specific IgE Test by Kit (D7450), Allergen-Specific IgE (D7451), and Allergen-Specific IgE by MAST (D7460)” is not explained. Such data must be clarified before publication.

Supplementary Table 2: This is an international journal, and the codes alone are meaningless to the readers. More context and explanations are necessary.

Conclusion: The conclusion is too general and lacks specificity. It should be more precise, directly summarizing the study's findings. Currently, it does not convey clear conclusions.

Author Response

Comments 1: Title: The current title is unclear; it should explicitly mention “Traditional Korean Medicine…” to accurately reflect the study’s focus.

Response 1: Thank you for your valuable suggestion. We have incorporated your feedback by adding "traditional" to the title (Line 2).

Comments 2: Figures: The figures need more detailed explanations. Currently, they only have titles without legends or descriptions. Each abbreviation used in a figure must be defined in the legend, even if it has been introduced in the main text.

Response 2: Following your advice, we have added detailed abbreviations and explanations to the figures and tables.

Figure 1: figure legends in line 100-103

Table 1: abbreviation of the table in line 185

Figure 2: figure legends in line 198-199

Comments 3: Supplementary Table: As indicated by its name, the supplementary table should be provided as a separate file.

Response 3: Following your advice, we have submitted the supplementary table as a separate file.

Comments 4: Supplementary Table 1: The data in this table is unclear. The distinction between “Allergen-Specific Immunoglobulin-IgE (D744001), Allergen-Specific IgE Test by Kit (D7450), Allergen-Specific IgE (D7451), and Allergen-Specific IgE by MAST (D7460)” is not explained. Such data must be clarified before publication.

Response 4: Regarding Supplementary Table 1, under the national health insurance reimbursement system, the same test may have different unit costs depending on the specific allergen tested, resulting in separate codes. Therefore, to ensure comprehensive extraction of the same test, we reviewed all relevant codes.

Comments 5: Supplementary Table 2: This is an international journal, and the codes alone are meaningless to the readers. More context and explanations are necessary.

Response 5: For the diagnostic codes used in Supplementary Table 2, we have provided a more detailed explanation at the bottom of the table.

Comments 6: Conclusion: The conclusion is too general and lacks specificity. It should be more precise, directly summarizing the study's findings. Currently, it does not convey clear conclusions.

Response 6: We have revised the conclusion section in accordance with the reviewer's comment,

Line 347-352: Despite methodological constraints, our large-population analysis provides significant real-world evidence on treatment patterns in Korea. The observed potential benefits of combining conventional medicine with KM warrant further investigation through more rigorous research designs across diverse populations. By addressing existing gaps in knowledge and practice, this study contributes to advancing patient-centered, evidence-based approaches in pediatric healthcare.

Reviewer 4 Report

Comments and Suggestions for Authors

Authors exploit data from a large Korean database of national insurance. They include several hundreds thousands children with allergic rhinitis (AR) that have visited a conventional western medicine hospital and a traditional Korean medicine provider. Some of their conclusions are that older children and those with comorbidities tend to use both services more than the rest of the patients. A positive point in their methodology is the selection process they have utilized, as depicted in figure 1, that increases the chances of correct diagnosis for allergic rhinitis. Nevertheless I have some comments

Major comments

Authors do not specify the timing of the visit in a Korean Medicine (KM) provider. Maybe a visit on KM provider between the two hospital visits would be an indication of a real concurrent treatment. Otherwise a visit on KM provider prior to the hospital may indicate a failure of KM. Thus lines 210-212 or 216-219 cannot be supported  

Comorbid rhinitis? It is not clear what comorbid rhinitis indicates? children diagnosed also with another form of rhinitis in addition to allergic rhinitis?

Minor comments

The cost of 512.7 billion KRW (line 43) should also be written in USD (in parenthesis) since the manuscript is of global interest and not only of topical (national) value.

I suppose in the schematic flow chart (figure 1) that children with cancer etc who have been excluded were 15,992 and not 582000.

Author Response

Major comments

Comments 1: Authors do not specify the timing of the visit in a Korean Medicine (KM) provider. Maybe a visit on KM provider between the two hospital visits would be an indication of a real concurrent treatment. Otherwise a visit on KM provider prior to the hospital may indicate a failure of KM. Thus lines 210-212 or 216-219 cannot be supported  

Response 1: In this study, valid KM treatment was defined as at least two visits to a KM hospital or clinic during the observation period. To ensure clarity, we have explicitly stated this criterion as follows.

Line 141-143: “To clearly define traditional KM combination therapy, only patients who received at least two separate KM treatment sessions during the observation period were included in the study.”

Comments 2: Comorbid rhinitis? It is not clear what comorbid rhinitis indicates? children diagnosed also with another form of rhinitis in addition to allergic rhinitis?

Response 2: Thank you for pointing out this ambiguity. The term "comorbid rhinitis" used in our analysis referred specifically to chronic rhinitis or chronic nasopharyngitis diagnosed alongside allergic rhinitis. In response to your valuable comment, we have revised both the main text and Figure 2 accordingly to clearly indicate "chronic rhinitis/chronic nasopharyngitis." We appreciate your insightful suggestion to enhance clarity in our manuscript.

Line 203-204: “Additionally, patients with chronic rhinitis/nasopharyngitis (OR: 1.94, 95% CI: 1.31 to 2.86)

Table 2 / Figure2

Minor comments

Comments 3: The cost of 512.7 billion KRW (line 43) should also be written in USD (in parenthesis) since the manuscript is of global interest and not only of topical (national) value.

Response 3: We have revised the sentence in accordance with the reviewer's comment.

Line 44-45: medical expenses for AR treatment reached 345 million USD (512.7 billion KRW)

Comments 4: I suppose in the schematic flow chart (figure 1) that children with cancer etc who have been excluded were 15,992 and not 582000.

Response 4: Thank you for your careful review and comment. Regarding the schematic flow chart (Figure 1), we confirm that there is no numerical error. To clarify, patients were excluded sequentially: first, individuals with cancer, cerebrovascular disease, or renal failure (n=582,798) were excluded. Subsequently, from the remaining subjects, those with less than two hospital visits during the observation period (n=15,992) were further excluded. Thus, the numbers presented in Figure 1 accurately reflect this sequential exclusion process. We appreciate your attention to detail, and we will make sure this is clearly stated in the manuscript.

Round 2

Reviewer 2 Report

Comments and Suggestions for Authors

Most of the issues were addressed.

Author Response

Comment 1: Most of the issues were addressed.

Response 1: Thank you for your positive feedback and for acknowledging that we have addressed most of the issues raised. We greatly appreciate your constructive suggestions, which have significantly improved the clarity and quality of our manuscript.

Reviewer 3 Report

Comments and Suggestions for Authors

The authors have answered the comments, which I believe improved the manuscript.

Author Response

Comment 1: The authors have answered the comments, which I believe improved the manuscript.

Response 1: Thank you for your positive feedback and for acknowledging that we have addressed most of the issues raised. We greatly appreciate your constructive suggestions, which have significantly improved the clarity and quality of our manuscript.

Reviewer 4 Report

Comments and Suggestions for Authors

dear authors response to my first comment is not clear for me

Author Response

Comment 1: dear authors response to my first comment is not clear for me
Response 1: Thank you for pointing out that our previous response was unclear, and we apologize for any confusion. 

Previous Comment 1: Authors do not specify the timing of the visit in a Korean Medicine (KM) provider. Maybe a visit on KM provider between the two hospital visits would be an indication of a real concurrent treatment. Otherwise a visit on KM provider prior to the hospital may indicate a failure of KM. Thus lines 210-212 or 216-219 cannot be supported  

Response: Thank you for your valuable comment. We understand the importance of considering the timing of KM visits in relation to conventional treatment and have revised the manuscript to clarify our criteria. In this study, we applied a one-year washout period (January to December 2017) to ensure that only newly diagnosed pediatric AR patients from 2018 onward were included. To define traditional KM combination therapy, we classified patients as receiving combination therapy only if they had a cross-visit between KM and conventional medicine within 10 days of their initial diagnosis. Furthermore, we ensured treatment continuity by including only patients who had a follow-up visit within 90 days of their previous visit. Finally, to confirm the receipt of KM treatment, patients were included in the study only if they had at least two separate KM visits for the same diagnosis during the observation period. We have explicitly stated these criteria in the manuscript to enhance clarity and address the concern regarding the timing of KM visits.

Line 148-156: “To clearly define traditional KM combination therapy, we applied a one-year washout period (January to December 2017) and included only pediatric patients newly diagnosed with AR from 2018 onward. Combination therapy was defined as cases where patients had a cross-visit between KM and conventional medicine (WM) within 10 days of the initial diagnosis. Additionally, treatment continuity was ensured by including only patients who had a follow-up visit within 90 days of their previous visit. To confirm the receipt of KM treatment, only patients who had at least two separate KM visits for the same diagnosis during the observation period were included in the study.”